

# 20 m Annual Paddy Rice Map for Mainland Southeast Asia Using Sentinel-1 SAR Data

Chunling Sun[1,2,3], Hong Zhang[1,2,3*], Lu Xu[1,2], Ji Ge[1,2,3], Jingling Jiang[1,2,3], Lijun Zuo[1,2], Chao Wang[1,2,3]

[1]Key Laboratory of Digital Earth Science, Aerospace Information Research Institute, Chinese Academy of Sciences, Beijing 100094, China
[2]International Research Center of Big Data for Sustainable Development Goals, Beijing 100094, China
[3]College of Resources and Environment, University of Chinese Academy of Sciences, Beijing 100049, China

*Correspondence to*: Hong Zhang (zhanghong@radi.ac.cn)

**Abstract.** Over 90% of the world's rice is produced in the Asia-Pacific Region. Synthetic aperture radar (SAR) enables all-day and all-weather observations of rice distribution in tropical and subtropical regions. Rice growth patterns in tropical and subtropical regions are complex, and it is difficult to construct representative rice growth patterns, which makes it much more difficult to extract rice distribution based on SAR data. To address this problem, a rice mapping method based on time-series Sentinel-1 SAR data is proposed in this study for large regional tropical or subtropical areas. Based on the analysis of rice backscattering characteristics in mainland Southeast Asia, the combination of spatio-temporal statistical features with the generalization ability to complex rice growth patterns was selected, then input into the U-Net semantic segmentation model and combined with WorldCover data to eliminate false alarms, and finally the 20-meter resolution rice map of five countries in mainland Southeast Asia in 2019 was obtained. On the validation sample set, the proposed method achieved an accuracy of 92.20%. Good agreement was obtained when comparing our rice map with statistical data and other rice maps at the national and provincial levels. The maximum coefficient of determination $R^2$ was 0.93 at the national level and 0.97 at the provincial level. These results demonstrate the advantages of the proposed method in rice extraction with complex cropping patterns and the reliability of the generated rice maps. The 20 m annual paddy rice map for mainland Southeast Asia is available at https://doi.org/10.5281/zenodo.7315076(Sun, 2022).

## 1 Introduction

Among the 17 sustainable development goals (SDGs) set by the United Nations in 2015, eradicating hunger, achieving food security and improving nutrition, and promoting sustainable agriculture are key components of Goal 2 "Zero Hunger"(Desa, 2016). Rice feeds more than half of the world's population as a staple food and is a major crop for world food security (Kuenzer and Knauer, 2012). Asia is the largest rice-producing region in the world (Chen et al., 2012). With 48 million hectares under rice cultivation, Southeast Asia accounts for 40% of global rice exports(Yuan et al., 2022). The dual pressure of population and environment threatens the sustainability of global food security(Faostat, 2010; Godfray et al., 2010). Accurate information





on planted area and spatial distribution is the basis for monitoring rice growth and predicting yield(Mosleh et al., 2015; Laborte et al., 2017; Clauss et al., 2018; Jin et al., 2018; Yu et al., 2020; Hoang-Phi et al., 2021).

Remote sensing technology plays a crucial role in rice growth monitoring and distribution mapping (Weiss et al., 2020; Zhao et al., 2021; Tsokas et al., 2022). Rice area mapping at the national scale usually uses medium- and low-resolution optical remote sensing data, such as MODIS and Landsat data. Some researchers used MODIS multitemporal data to produce rice maps of China with resolutions of 500 m, 250 m and 500 m respectively (Sun et al., 2009; Clauss et al., 2016; Qiu et al., 2022). Guan et al. produced rice maps of Vietnam at 500 m resolution using MODIS time series data in 2010(Guan et al., 2016). The National Agricultural Statistics Service (NASS) released the state-based Crop Data Layer (CDL), a 30-m resolution crop distribution map product for the entire continental United States, using multisource medium resolution remote sensing data (Landsat, IRS-p6, DEIMOS-1, etc.) (Johnson and Mueller, 2010).Luo et al. and Wei et al. used Landsat time-series data to produce 1 km and 30 m resolution rice datasets for China, respectively (Luo et al., 2020a; Luo et al., 2020b; Wei et al., 2022). Recently,the Sentinel -2 satellite sensor opens up new possibilities for paddy rice monitoring. Liu et al. obtained medium-resolution rice maps of China using Sentinel-2 time series data in 2020 (Liu et al., 2022).

At the continental scale, MODIS time-series data were frequently used to map the distribution of rice cultivation (Dong et al., 2016a; Dong et al., 2016b). Xiao et al. and Gumma et al. produced low- and medium-resolution rice area maps for several South and Southeast Asian countries using MODIS data at the 500 m spatial resolution, respectively (Xiao et al., 2005; Xiao et al., 2006; Gumma et al., 2011a; Gumma et al., 2011b; Bridhikitti and Overcamp, 2012; Gumma et al., 2014; Nelson and Gumma, 2015). Using MODIS time-series data, Zhang et al. generated rice acreage maps for China and India from 2000 to 2015 (Zhang et al., 2017). Han et al. used MODIS data to complete 500 m annual rice maps for the Asian monsoon region from 2000 to 2020(Han et al., 2022). SPOT data were also used for continent-wide rice mapping. Manjunath et al. used 2009-2010 multi-temporal SPOT VGT normalized difference vegetation index (NDVI) data to produce 1km resolution rice maps for South and Southeast Asia(Manjunath et al., 2015).

Most of the rice in the world is distributed in hot and rainy areas. However, optical data are easily obscured by clouds, which also poses a challenge for rice extraction in humid and sub-humid climates with abundant water resources such as Southeast Asia(Zhu and Woodcock, 2012; Liu et al., 2019; Sun et al., 2021). Compared with traditional optical remote sensing, synthetic aperture radar (SAR) is an active microwave radar with the advantages of all-day and all-weather, which is weather-independent and can penetrate clouds, and is very sensitive to the geometric structure and dielectric properties of crops(Huang et al., 2017; Orynbaikyzy et al., 2019; Sun et al., 2022). In recent years, free SAR data represented by Sentinel-1 data have been widely used in rice mapping over large regions. Singha et al. obtained seasonal rice maps at 10 m resolution for Bangladesh and northeast India using time-series Sentinel-1VH data for 2017 (Singha et al., 2019). Pan et al. used 2016-2020Sentinel-1VH data to produce 10-m spatial resolution double-season rice maps for nine provinces in southern China (Pan et al., 2021). Xu et al. used time-series Sentinel-1VH data to obtain a 20 m rice map for Thailand in 2019 (Xu et al., 2021).

To take full advantage of multi-source remote sensing data, some researchers combined optical and SAR time-series data in the large-scale rice mapping studies (Thenkabail et al., 2009; Zhang et al., 2018; You and Dong, 2020). Phan et al. used





Sentinel 1/2 and Landsat data to produce the first Vietnam land use/land cover annual dataset with 30m resolution from 1990 to 2020 (Phan et al., 2021). Han et al. obtained 500m resolution rice maps from 2017 to 2019 in Northeast and Southeast Asia using Sentinel-1 and MODIS time-series data (Han et al., 2021).

At present, large-scale rice mapping methods based on remote sensing data can be divided into two categories, one is the combination of phenological information and remote sensing images, and the other is the combination of time series data and

machine learning relying on image information. The phenology-based approach refers to the extraction of rice by defining phenological indicators or identifying phenological periods by combining the time-series data of the rice growth cycle and the analysis of the growth phenology calendar (Nelson et al., 2014; Chen et al., 2016; Nguyen and Wagner, 2017; Liu et al., 2018; Xin et al., 2020; Ni et al., 2021). The phenological periods such as transplanting, flooding, heading and maturity are most often used to extract rice. Shew et al. combined vegetation indices extracted from Landsat time-series data with a rule-based

algorithm for phenological stages to map a 30 m dry season rice map of Bangladesh from 2014 to 2018 (Shew and Ghosh, 2019). Li et al. extracted the minimum and maximum values of permanent water backscatter coefficients and three thresholds of phenological characteristics, namely, the date of the beginning of the season, date of maximum backscatter during the peak growing season, and length of the vegetative stage from 402 scenes of Sentinel-1 data in 2017 to map rice paddies in the Mun River basin ,Thailand (Li et al., 2020).Kang et al. completed a 10 m resolution rice map of Cambodia from Sentinel-1 (2015)

and Sentinel-2 (2015-2017) time-series data using three key rice phenological periods in the dry and rainy seasons, respectively (Kang et al., 2022).

However, the phenology-based methods rely too much on human intervention and are not suitable for rice extraction with complex cropping cycles. The approaches based on the combination of time series data and machine learning method refer to the direct use of time series as the input features for machine learning (Ndikumana et al., 2018; Chang et al., 2020; Mansaray

et al., 2021; Yang et al., 2021). Machine learning methods are used to extract rice information by mining fixed relationships across growth periods of rice (Yang et al., 2019; You et al., 2021). Torbick et al. used Sentinel-1, Landsat-8 and PALSAR-2 time series data and a random forest algorithm to map rice planting area and planting intensity of Myanmar with 20 m resolution in 2015 (Torbick et al., 2017). Inoue et al. developed a 30 m resolution map of paddy rice in Japan for 2018 using Sentinel-1 SAR data and Sentinel-2 data with the conventional decision tree methods (Inoue et al., 2020). Wei et al. completed rice area

mapping for the Arkansas River Basin, USA, by entering dual-polarized Sentinel-1 data from 2017-2019 into a modified U-Net model (Wei et al., 2021). Soh et al. used Sentinel-1 and Sentinel-2 time series data and a K-means clustering method to map rice in West Malaysia (Soh et al., 2022).

The climate in tropical or subtropical regions such as Southeast Asia is suitable for rice growth throughout the year, increasing the difficulty of rice extraction. First, it is difficult to obtain accurate phenological information because the climate in Southeast

Asia is hot and humid for rice growth, the timing of rice seedling and transplanting is more flexible (Xu et al., 2021). This makes it impossible to use empirical methods to determine effective weathering indicators and suitable periods. Second, rice growth patterns in Southeast Asia are too complex to construct a representative rice growth model (Kang et al., 2022). This poses obstacles for rice extraction methods that utilize time-fixed relationships in time-series data.

Current publicly downloadable rice products for Southeast Asia include the Asia rice map (IRRI Rice Data, 500 m) (Nelson
and Gumma, 2015), Vietnam-wide annual land use/land cover datasets from 1990 to 2020 (VLUCD, 30 m) (Phan et al., 2021),
annual paddy rice maps for Northeast and Southeast Asia from 2017 to 2019 (NESEA-Rice10, 10 m) (Han et al., 2021), and
annual rice in the Asian monsoon region from 2000 to 2020 (500 m) (Han et al., 2022). Except for Vietnam's VLUCD, the
source data for the public rice maps in Southeast Asia were mainly MODIS. Rice maps using MODIS data contained a large
number of mixed pixels due to low spatial resolution (Dong et al., 2015; Shew and Ghosh, 2019), which affected the accuracy
of rice maps.

Therefore, in this study, to meet the requirements of high-precision rice area mapping inSoutheast Asia, the objectives
accomplished using Sentinel-1 time-series data are as follows.

(1)    A new feature extraction method is proposed by analyzing the time-series backscattering variation of rice in mainland
Southeast Asia. The method does not need to summarize the general evolutionary model from rice backscatter coefficients
with diverse growth patterns. Using three simple but effective temporal statistical features defined in this study, it is
possible to capture features that provide key information about the rice growth process. This study provided a new idea
for rice mapping methods in tropical or subtropical regions.

(2)    A deep combination of the above features and the U-Net model will be used to fully exploit the pixel-level semantic
features to complete the annual rice mapping of five Southeast Asian countries in 2019, enriching the available Southeast
Asian rice maps, and providing support information for the scientific community and scientific decision-making.

The rest of the paper is organized as follows. Section 2 describes the study area and the data information used; Section 3
presents the rice mapping scheme; Section 4 presents the temporal characteristics analysis of the data and the rice map results;
Section 5 discusses the results; and finally, Section 6 draws conclusions.

## 2 Materials

### 2.1 Study area


Approximately 90% of the world's rice is grown on 140 million hectares of land in Asia. The rice production in mainland
Southeast Asian accounts for about 15% of the world rice production(Fao, 2020).. The study area is five countries in mainland
Southeast Asia, namely, Myanmar, Thailand, Laos, Cambodia, and Vietnam, as shown in Figure 1. These countries have more
land under rice cultivation than any other crop. And Vietnam and Thailand are the two largest rice exporters in the world (Yuan
et al., 2022). Indeed, changes in rice production in these countries could destabilize international rice markets and have a clear
impact on global food security.

Southeast Asia has a tropical monsoon climate with an average annual temperature of 20-27 °C and abundant rainfall.
Therefore, rice can be grown at any time of the year. Agricultural systems in Southeast Asia are dominated by rainfed lowland
rice and irrigated lowland rice (Kuenzer and Knauer, 2012). Under suitable irrigation conditions, rice can be harvested two to
three times per year.

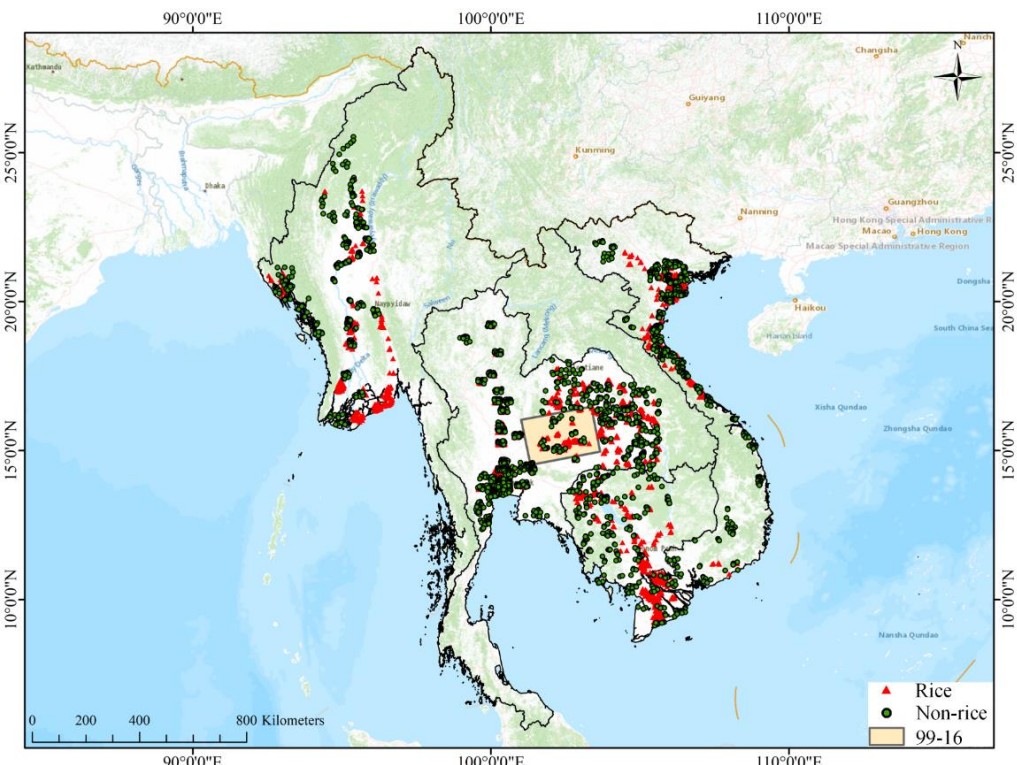

**Figure 1: Location of the study area. Orbit-frame 99-16 images are used for the training samples, and the Rice and Non-rice flags**
**show the distribution of the validation sample set. The base map is from Esri.**

### 2.2 Data source

### 2.2.1 Satellite imagery and auxiliary data

The European Space Agency (ESA) provides a free data source for large-scale land cover monitoring through Sentinel-1A, launched in 2014, and Sentinel-1B, launched in 2016 (Torres et al., 2012). The Sentinel-1 satellites carry a C-band (5.405 GHz)
synthetic-aperture radar with a 12-day revisit period. In this study, the 2019 dual-polarized (VV/VH) GRD products in Interferometric Wide Swath (IW) mode were downloaded from the ASF website. In total, 12 tracks, 90 frames and 2665 scenes of data were acquired. Details are shown in Table 1.

And, the DEM and land use/land cover product were also collected. Shuttle Radar Topography Mission (SRTM) 3sec DEM product was used for terrain correction of SAR data. WorldCover data were used to reduce false alarms caused by water and
woodland. WorldCover is a global land cover product produced by ESA and several scientific institutions using Sentinel-1 and -2 data (Zanaga et al., 2021). It provides information on 11 land cover types for 2020 with a resolution of 10 m and an overall accuracy of 80.7% for the Asian region.





**Table 1 List of SAR data**

| Country | Orbit | Frame | Satellite | Number of images | Country | Orbit | Frame | Satellite | Number of images |
|---|---|---|---|---|---|---|---|---|---|
| | | | | Experimental Data | | | | | |
| Thailand | 172 | 17 | S1A | 31 | Laos | 99 | 1240 | S1A | 30 |
| | | 18 | S1A | 31 | | | 1245 | S1A | 30 |
| | 135 | 16 | S1A | 23 | | | 1250 | S1A | 30 |
| | | 17 | S1A | 23 | | 26 | 44 | S1A | 31 |
| | | 18 | S1A | 23 | | | 49 | S1A | 31 |
| | | 19 | S1A | 23 | | | 54 | S1A | 31 |
| | 62 | 1 | S1B | 29 | | | 59 | S1A | 31 |
| | | 2 | S1B | 29 | | | 64 | S1A | 31 |
| | | 3 | S1B | 29 | | | 69 | S1A | 31 |
| | | 4 | S1B | 29 | | 128 | 44 | S1A | 30 |
| | | 5 | S1B | 29 | | | 49 | S1A | 30 |
| | | 20 | S1A | 27 | | | 54 | S1A | 30 |
| | | 21 | S1A | 27 | | | 59 | S1A | 30 |
| | | 22 | S1A | 26 | | | 64 | S1A | 30 |
| | | 23 | S1A | 24 | Myanmar | 41 | 44 | S1A | 31 |
| | | 24 | S1A | 25 | | | 50 | S1A | 31 |
| | 164 | 1 | S1B | 32 | | | 55 | S1A | 31 |
| | | 2 | S1B | 32 | | | 60 | S1A | 31 |
| | | 3 | S1B | 32 | | | 65 | S1A | 31 |
| | | 4 | S1B | 32 | | | 70 | S1A | 31 |
| | | 5 | S1B | 32 | | 143 | 46 | S1A | 30 |
| | | 20 | S1A | 13 | | | 51 | S1A | 30 |
| | 91 | 1 | S1B | 32 | | | 56 | S1A | 30 |
| | | 2 | S1B | 32 | | | 61 | S1A | 30 |
| | | 3 | S1B | 32 | | | 66 | S1A | 30 |
| | | 4 | S1B | 32 | | | 71 | S1A | 30 |
| Cambodia | 99 | 1220 | S1A | 30 | | | 76 | S1A | 30 |
| | | 1225 | S1A | 30 | | 70 | 1217 | S1A | 31 |
| | | 31 | S1B | 31 | | | 1222 | S1A | 31 |
| | 26 | 29 | S1A | 28 | | | 1227 | S1A | 31 |
| | | 32 | S1B | 30 | | | 1232 | S1A | 31 |
| | | 38 | S1B | 30 | | | 1237 | S1A | 31 |
| | | 43 | S1B | 30 | | | 1242 | S1A | 31 |
| | 128 | 34 | S1A | 30 | | | 1247 | S1A | 31 |
| | | 39 | S1A | 30 | | | 1252 | S1A | 31 |
| Vietnam | 26 | 23 | S1A | 28 | | | 1257 | S1A | 31 |
| | | 34 | S1A | 31 | | | 1262 | S1A | 31 |
| | 55 | 31 | S1A | 31 | | | 1267 | S1A | 31 |
| | | 37 | S1A | 31 | | 172 | 1248 | S1A | 28 |
| | | 42 | S1A | 31 | | | 1253 | S1A | 28 |
| | | 47 | S1A | 31 | | | 1258 | S1A | 28 |
| | | 62 | S1A | 31 | | | 1263 | S1A | 28 |
| | | 67 | S1A | 31 | | | 1268 | S1A | 28 |
| | | 72 | S1A | 31 | | | 1273 | S1A | 28 |
| | 128 | 29 | S1A | 30 | | | | | |
| | | 69 | S1A | 30 | | | | | |
| | | | | Training data set | | | | | |
| Thailand | 99 | 16 | S1A | 29 | | | | | |



### 2.2.2 Agricultural statistics

The statistical yearbooks of each country were collected to compile annual census data of rice harvested area at different administrative levels in these countries. The administrative levels include national and subnational levels (state, province, or regions, uniformly represented by province in this study). The unit of area in the statistical data is uniformly converted to hectares (ha).

### 2.2.3 Available rice maps based on remote sensing data

From the perspective of resolution and coverage area, 2 publicly downloadable rice maps were selected for comparison.

(1) Vietnam-wide annual land use/land cover datasets (VLUCD)

Researchers from the Japan Aerospace Exploration Agency (JAXA) produced the first 30-m resolution Vietnam-wide annual land use/land cover datasets (VLUCD) using multiple sources of data (including Landsat and Sentinel-1/2) and a random forest algorithm (Phan et al., 2021). The VLUCD contains annual land cover products for 1990-2020, including a primary

classification (containing 10 different categories of primary land cover, including rice) and a secondary classification (18 different categories of secondary primary land cover, including rice). The rice layer was extracted from the 2019 annual land cover products for comparison.

(2) Rice data of Asia from International Rice Research Institute (IRRI Rice Data)

The International Rice Research Institute (IRRI) is an international agricultural research and training organization with its

headquarters in Los Baños, Laguna, in the Philippines, and offices in seventeen countries. IRRI is one of 15 agricultural research centers in the world that form the Consortium of International Agricultural Research Centers(CGIAR), a global partnership of organizations engaged in research on  food security. IRRI is also the largest non-profit agricultural research center in Asia.

The IRRI produced a 500 m resolution map of the general distribution of rice in Asia from 2001 to 2012 using MODIS time-

series data (Nelson and Gumma, 2015), which is freely available to the public.

Table 2 shows details of the SAR data, auxiliary data, available rice maps, land cover data and statistics used in the study.

**Table 2 Details of the data used in the study**

| Data type | Data product or country name | Year | Resolution | Description | Data access | Last access (dd/mm/yyyy) |
|---|---|---|---|---|---|---|
| SAR imagery | Sentinel-1 | 2019 | 20*22 m (rg*az) | The backscatter characteristics extraction | https://search.asf.alaska.edu/#/ | 11/10/2022 |
| DEM | Shuttle Radar Topography | 2000 | 90m | Terrain correction | https://search.earthdata.nasa.gov/search?q=SRTM | 11/10/2022 |



| | | | | | | |
|---|---|---|---|---|---|---|
| | Mission (SRTM) 3sec | | | | | |
| Land cover data | ESA WorldCover 2020 | 2020 | 10 m | Extraction of water and forest mask | https://esa-worldcover.org/en | 11/10/2022 |
| Available Rice Map | Vietnam-wide annual land use/land cover datasets (VLUCD) | 2019 | 30 m | Spatial consistency comparison of rice extraction results | https://www.eorc.jaxa.jp/ALOS/en/dataset/lulc/lulc_vnm_v2109_e.htm | 11/10/2022 |
| | Rice data of Asia from IRRI (IRRI Rice Data) | 2000 to 2012 | 500 m | Spatial consistency comparison of rice extraction results | https://www.irri.org/mapping | 11/10/2022 |
| Statistical yearbook | Vietnam | 2019 | Province scale | verifying the classification accuracy | https://www.gso.gov.vn/en/homepage/ | 11/10/2022 |
| | Cambodia | 2019 | Province scale | verifying the classification accuracy | http://nis.gov.kh/index.php/km/ | 11/10/2022 |
| | Laos | 2019 | Province scale | verifying the classification accuracy | https://www.lsb.gov.la/en/home/ | 11/10/2022 |
| | Thailand | 2019 | Province scale | verifying the classification accuracy | http://www.nso.go.th/sites/2014en | 11/10/2022 |
| | Myanmar | 2019 | State scale | verifying the classification accuracy | https://www.mopf.gov.mm/en/page/planning/central-statistical-organization-cso/752 | 11/10/2022 |

## 3 Method

The flowchart of this study is shown in Figure 2. First, the Sentinel-1 time series images were preprocessed. Then, key features in the rice growth process are extracted from the time series SAR data. To make full use of the pixel-level semantics of the features, the extracted features were fed into the U-Net model to obtain rice extraction results with spatial details. Finally, to reduce false alarms from water bodies and vegetation, the results were postprocessed using masks generated based on high-precision land cover products to obtain the annual rice map of five Southeast Asian countries.

**3.1 Preprocessing**

The Sentinle-1 time-series data were preprocessed using SNAP software (Filipponi, 2019). The steps were as follows: (1) orbit correction; (2) thermal noise removal; (3) radiometric calibration; (4) coregistration; (5) terrain correction; (6) multitemporal



speckle noise filtering; and (7) conversion of the multitemporal intensity map to sigma 0 ($\sigma^0$) on the decibel (dB) scale. The final $\sigma^0$ images with 20 m resolution in the WGS84 geographic coordinate system were obtained.

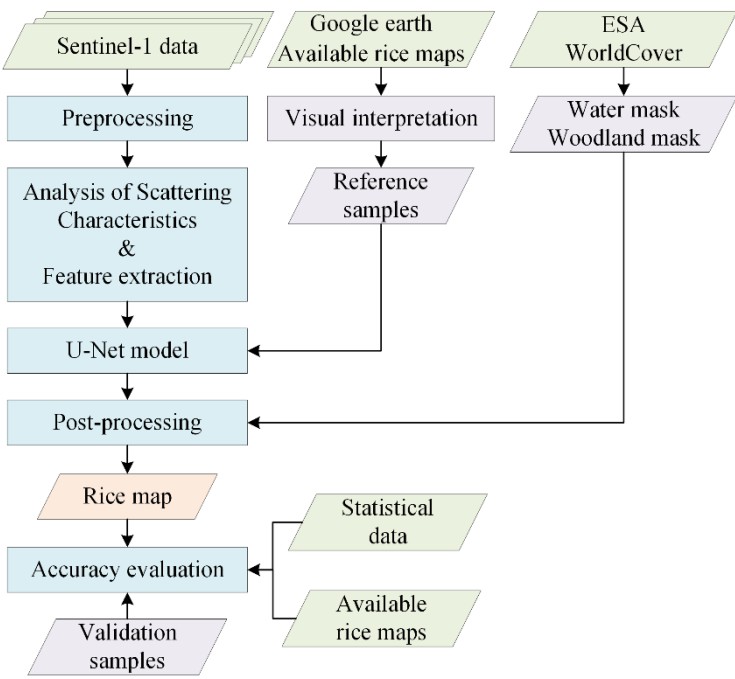


**Figure 2: Flowchart of the proposed rice mapping method using Sentinel-1 data.**

## 3.2 Feature Extraction

As described in many previous studies (Singha et al., 2019; Chang et al., 2020; Crisóstomo De Castro Filho et al., 2020; Sun et al., 2022), VH polarization was more sensitive to the flooding period of rice than VV polarization and has been more widely
used for rice extraction. Therefore, VH data were selected in this study. To analyze the time-series characteristics of the backscattering coefficients of rice and other land cover types in the study area, representative sample plots of four typical land cover types (rice, water bodies, buildings, and vegetation) were selected. Based on Google Earth data and other land cover data, four rice regions that belongs to different cropping systems were chosen. The average VH polarization time series data of these land cover types were calculated, as shown in Figure 3.
In Figure 3, the backscattering coefficients of water bodies were small, as they exhibited single specular scattering, and their return power level was lower than other land covers. In contrast, buildings exhibited double bounce and their return powers were much stronger, leading to larger backscattering coefficients. The scattering process of radar waves of vegetation was more complicated, and the backscattering coefficients of vegetation were between buildings and water bodies. For different kinds of rice samples, the curve fluctuations were significant, due to the effects of flooding and the multi-season planting
patterns. But generally, their backscattering intensities ranged between buildings and water bodies. More specifically, the land parcel of Rice 1 was planted with two seasons of rice during the observation period: the first season was from April to July,

and the second season was from August to October. The land parcel of Rice 2 was planted with only one season of rice, from April to September. The land parcel of Rice 3 was planted with two seasons of rice: the first season was from March to July and the second season was from July to October. The land parcel of Rice 4 was planted with three seasons of rice: the first

season was from February to June, the second season was from June to October, and only part of the third season (October-December) was observed. It can be seen that the time steps of each growing season for the selected Rice 1- Rice 4 were inconsistent. In fact, the high heterogeneity of rice backscattering coefficients in Southeast Asia is caused by the high heterogeneity in climate and topography. This makes the backscatter coefficient curves of the rice growth cycle more diverse and does not allow us to summarize a generalized model of rice evolution. Therefore, it will be difficult to accomplish the rice

extraction task using a direct reliance on the fixed relationship between phenology and time.

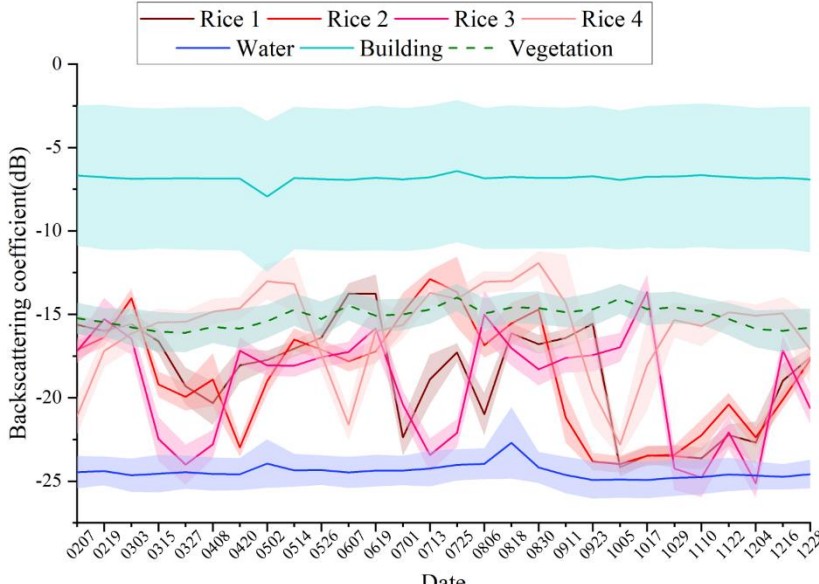

**Figure 3: The average VH polarization backscattering coefficient curve of typical landcovers**

Through a large number of comparative experimental analysis and combined with our previous research work (Xu et al., 2022), three time-series statistical features that can describe the most significant SAR characteristics during rice growth were selected

for rice mapping in the study area, namely, the sharpness of the change in $\sigma^0$ ($\sigma^0_{var}$), the minimum value of the backscatter coefficients in the time-series ($\sigma^0_{min}$), the maximum value of backscatter coefficients in the time-series ($\sigma^0_{max}$).

The interaction between the crop canopy and microwave radiation varies with time during plant growth. In contrast, the backscattering coefficients of non-crops, such as water bodies, buildings and forest, are more stable. Therefore, the sharpness of the change in $\sigma^0$ with time will be a key factor in distinguishing cropland from other land cover types. $\sigma^0_{var}$ is given by the

following equation.



$$\sigma_{var}^0 = \frac{1}{n}\sum_i^n |\sigma_i^0 - \sigma_{mean}^0|^2 \tag{1}$$

where $\sigma_{mean}^0 = \frac{1}{n}\sum_i^n \sigma_i^0$, $n$ is the number of images.

The presence of a flooding period makes rice differ significantly from other crops in terms of backscattering characteristics. The backscattering coefficient of rice in the flooding period is close to that of the water body. Therefore, this study identified the flooding period by calculating the minimum value of the backscatter coefficient in the time-series images to distinguish rice from other crops. $\sigma_{min}^0$ is given by Equation (2).

$$\sigma_{min}^0 = min\{\sigma_1^0, \sigma_2^0, \sigma_3^0, \dots, \sigma_n^0\} \tag{2}$$

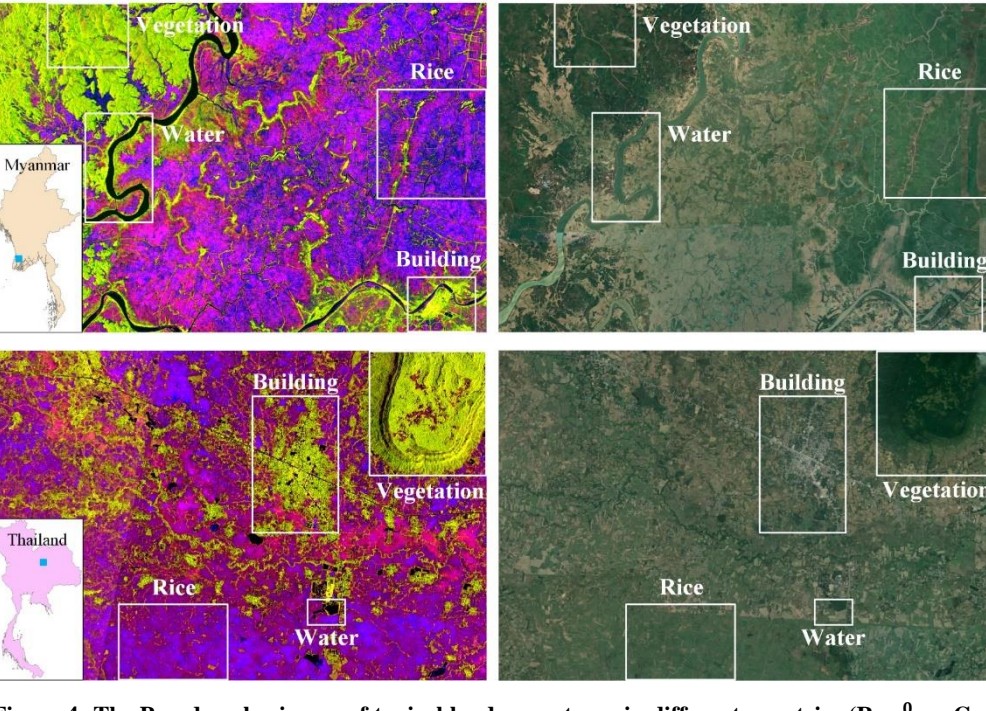

**Figure 4: The Pseudo-color image of typical land cover types in different countries (R: $\sigma_{max}^0$; G: $\sigma_{min}^0$; B: $\sigma_{var}^0$ ;Optical images are from Google Earth ©Google Earth).**



The seasonal backscattering variation exhibited by water bodies can interfere with the identification of rice. In contrast to the seasonal variation of water bodies, the backscatter coefficient of rice shows a substantial increase during the growth process. Therefore, false alarms generated by water bodies can be reduced by identifying the maximum value of backscatter coefficients in the time-series images. $\sigma_{max}^0$ is given by the following equation.

$$\sigma_{max}^0 = max\{\sigma_1^0, \sigma_{2,}^0 \sigma_3^0, ..., \sigma_n^0\} \quad\quad (3)$$

A pseudo-color image is synthesized in the order of R: $\sigma_{max}^0$, G: $\sigma_{min}^0$, and G: $\sigma_{var}^0$. False color images of typical land cover
for several countries and the corresponding optical images in Google Earth are given in Figure 4. Due to the higher $\sigma_{var}^0$ and $\sigma_{max}^0$ and lower $\sigma_{min}^0$ of rice, the color of rice in the pseudo-color composite image is mainly purplish red, sometimes red or dark blue. Compared to other land covers, water bodies have lower $\sigma_{var}^0$, $\sigma_{max}^0$ and $\sigma_{min}^0$. Therefore, water bodies are black in the false color composite. Land covers with less variation in backscatter intensity, such as buildings and vegetation, generally have smaller $\sigma_{var}^0$ and higher $\sigma_{min}^0$. Therefore, the colors of these land covers are usually yellow or green in the pseudo-color
image.

### 3.3 Training and validation sets

The above analysis shows that, the rice and non-rice landcovers of these Southeast Asian countries have consistent features in the pseudo-color image, which means that the model trained by one scene was applicable for all other scenes. Therefore, a
training dataset generated from the orbit-frame 99-16 images of Thailand from previous work (Xu et al., 2021) was used, as shown in Figure 1. A sliding window with a pixel size of 224*224 was used to slice the training images into image patches with 50%. The training dataset consisted of 15659 image patches with a pixel size of 224*224. A validation sample set for accuracy evaluation was collected using auxiliary data such as Google Earth optical images and other rice maps. The validation samples were divided into two categories, rice category and non-rice category, and the distribution is shown in Figure 1.
Specific information is shown in Table 3.

**Table 3 Validation sample set information**

| Class | Number of plots | Number of pixels |
|---|---|---|
| Rice | 1913 | 2,128,431 |
| Non-rice | 2032 | 2,188,477 |

### 3.4 U-Net Model

In this paper,high-resolution rice production was accomplished using U-Net. U-Net is a classical semantic segmentation
model widely used in biomedical image segmentation and remote sensing (Wei et al., 2019). It outputs semantically labeled



pixel-by-pixel images corresponding to the input image while extracting high-level semantic features, so that the spatial details of the input image can be maintained (Ronneberger et al., 2015).

The structure of U-Net model is shown in Figure 5. The model has 23 convolutional layers, including eighteen 3*3 convolutional layers, four 2*2 convolutional layers and one 1*1 convolutional layer. U-Net consists of encoder (contracting path) and decoder (expansive path). The encoder part consists of five downsampling units, where each unit consists of two 3*3 convolutional layers and a 2*2 max-pool layer. The output of the downsampling unit is input to the next downsampling unit by max-pooling. The decoder contains four upsampling units, each of which consists of two 3*3 convolutional layers and a 2*2 deconvolutional layer. In the final stage of the decoder, the feature vector of the last upsampling unitis converted into a probability mapping by the 1*1 convolutional layer. The dimension of the probability mapping is 2 and the pixel value indicates the probability that the pixel belongs to rice and non-rice.

To solve the problem of uneven data distribution, a batch normalization (BN) layer was added before each convolutional layer (Ioffe and Szegedy, 2015). The BN layer allows the input data to follow the same distribution to achieve regularization of the model.

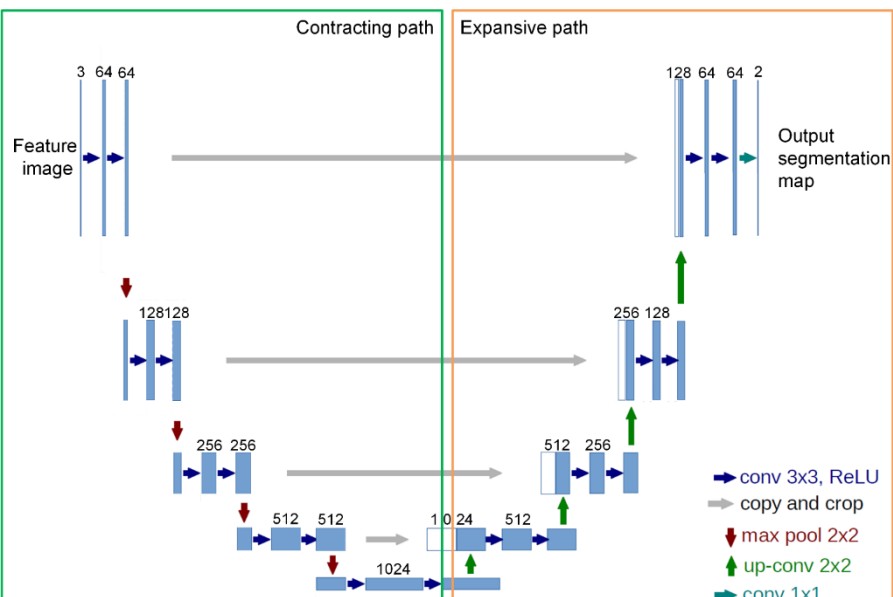

**Figure 5: Structure of the U-Net model.**

## 3.5 Postprocessing

In rice mapping, water bodies (e.g., rivers and lakes) can confuse the flooding period signal of rice. In addition, vegetation may cause some disturbances due to weather effects.

Therefore, as drawn on many studies (Lavreniuk et al., 2018; Cué La Rosa et al., 2019; Sun et al., 2021), water body masks and woodland masks produced by WorldCover were used to reduce false alarms of rice extraction results to some extent.



## 3.6 Accuracy evaluation

In this study, several strategies were used to evaluate our rice map product, including accuracy assessments based on validation sets and comparisons with statistical data and other rice maps at the national and provincial levels. First, common accuracy metrics based on the validation set were calculated to measure the classification effectiveness of the model, including accuracy, precision, recall, and kappa (Congalton, 1991; Vapnik, 1999; Mchugh, 2012).

$$Accuracy = \frac{TP + TN}{TP + TN + FN + FP} \tag{4}$$

$$Precision = \frac{TP}{TP + FP} \tag{5}$$

$$Recall = \frac{TP}{TP + FN} \tag{6}$$

$$Kappa = \frac{accuracy - P_e}{1 - P_e} \tag{7}$$

$$P_e = \frac{(TP + FP) \times (TP + FN) + (FN + TN) \times (FP + TN)}{(TP + TN + FN + FP)^2} \tag{8}$$

where *TP* denotes the number of pixels correctly classified as rice, *TN* denotes the number of pixels correctly classified as non-rice, *FP* denotes the number of pixels misclassified as rice among non-rice pixels, *FN* denotes the number of pixels misclassified as non-rice among rice pixels, and *P*e is the desired accuracy.

Second, the spatial consistency of rice extraction results with statistical data and other rice maps was compared at the national and provincial levels. The coefficient of determination ($R^2$) of the rice map with statistical data and other rice maps was calculated using the following equation (Draper and Smith, 1998).

$$R^2 = \frac{\left( \sum_{i=1}^{n} (x_i - \bar{x}_i) \times (k_i - \bar{k}_i) \right)^2}{\sum_{i=1}^{n} (x_i - \bar{x}_i)^2 \times \sum_{i=1}^{n} (k_i - \bar{k}_i)^2} \tag{9}$$

where $n$ is the total number of administrative units, $x_i$ is the area of extracted rice, $\bar{x}_i$ is its corresponding mean value, $k_i$ is the area of statistical data or other rice maps and $\bar{k}_i$ is its corresponding mean value.

## 4 Results

The 2019 rice map for mainland Southeast Asia using Sentinel-1 SAR data was shown in Figure 6. According to the extraction result, the main rice production areas in Myanmar are located in the Ayeyarwady, Bago and Yangon delta regions, which are crossed by river systems. In addition, Mandalay, Sagaing and Magwayue in the northern arid mountainous region also play an important role in rice production. Thailand's rice fields are concentrated in the central plains, north and northeast. The main rice-producing areas in Laos are located in the central and southern lowland areas. Many of the major rice-producing provinces are located along the Mekong River, such as Bolikhamxai, Khammouan, Savannakhet, Salavan, and Champasak. Rice fields

in Cambodia are concentrated in the Tonle Sap Lake basin and the southern Mekong River basin. The representative rice growing areas in Vietnam are the Mekong Delta and the Red River Delta.

Next, the rice map was evaluated as comprehensive as possible from three different scales. First, the validation sample set introduced in the previous section was used to evaluate the accuracy of rice mapping from the methodological level. Second, at the national level, the rice maps were compared with statistical data on rice harvested area and other available rice maps, respectively. Finally, at the provincial level, more detailed comparisons were made with statistical data and other provincial rice maps to measure the spatial consistency between the extracted rice distribution and these data.

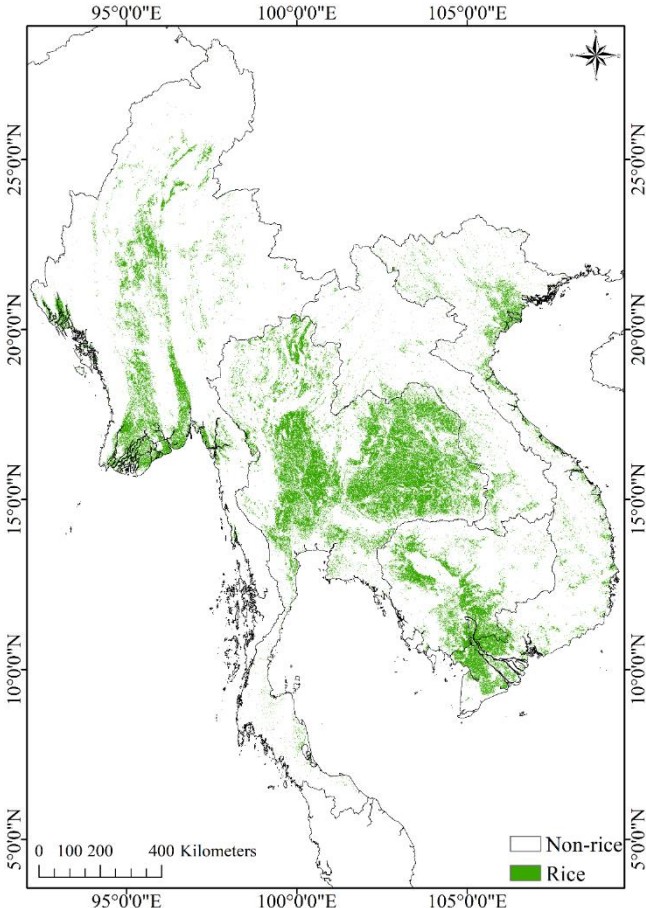

**Figure 6: 2019 20m resolution rice map of five countries in mainland Southeast Asia**

## 4.1 Accuracy based on the validation set

The accuracies of the rice map based on the validation sample set is shown in Table 4. Among them, the accuracy was as high as 92.20%, and the Kappa was 0.8425, which proved that the proposed method had good classification performance. The precision was 92.45%, which indicated that the method could effectively reduce the false alarms in the rice extraction results.



Therefore, these precision metrics illustrated that the rice mapping results were in good agreement with the validation samples.
It also further demonstrated the capability of the proposed method for rice mapping in large tropical regions.

**Table 4 Accuracy of the rice map based on the validation set**

| Class | Accuracy | Precision | Recall | Kappa |
|---|---|---|---|---|
| Rice | 92.20% | 92.45% | 90.26% | 0.8425 |

### 4.2 Comparison with statistical data and other rice maps at the national scale

Figure 7 showed the comparison of the extracted rice area with statistical data and the IRRI rice data at the national level scale
for five Southeast Asian countries. As seen from the figure, the extraction results were consistent with both statistical data and
IRRI rice data. Most points were distributed in the vicinity of the 1:1 line. In contrast, the extraction result was better consistent
with IRRI, $R^2$ can reach 0.93, while $R^2$ with statistical data was 0.78.

Compared with IRRI rice data, the extraction area of Cambodia, Laos and Thailand was close to that of IRRI, while that of
Myanmar and Vietnam was slight lower. Compared with the statistical data, the extraction areas of Cambodia and Laos were
in good agreement with the statistical data. The extraction area of rice in Myanmar and Vietnam was lower, while that in
Thailand was slightly higher.

Table 5 showed the statistical area of rice, the area of other rice maps and the area of rice extraction in Vietnam. It could be
seen that the statistics of rice harvested area were much higher than the area of rice extracted. The statistical data were the total
rice harvest areas in different growing seasons each year, but the extracted rice area was the land area where rice was planted.
In Vietnam, there are three seasons of rice, namely, spring rice, autumn rice and winter rice, while the harvested areas of spring
rice and autumn rice are comparable, and the harvested area of winter rice is smaller. In this way, part of the statistical data of
rice harvest area is repeated and accounts for a large proportion of the area, resulting in a larger rice statistical area than the
extracted rice area. Although other countries also have multiple rice seasons, the areas of rice in the main season are large,
while that in other seasons is small, so the area proportion calculated repeatedly is small. The extracted rice area was closer to
the paddy land area in statistical yearbook of Vietnam and VLUCD, indicating that the extraction result was reliable.

**Table 5 Statistics, other rice maps and the extracted rice area in Vietnam**

| Country | Statistics of rice cultivation area (*10^6ha) | IRRI rice data (*10^6ha) | Paddy Land area (*10^6 ha) | VLUCD (*10^6ha) | Extracted rice cultivation area (*10^6 ha) |
|---|---|---|---|---|---|
| Vietnam | 7.4695 | 6.1527 | 4.1205 | 3.8210 | 3.3270 |





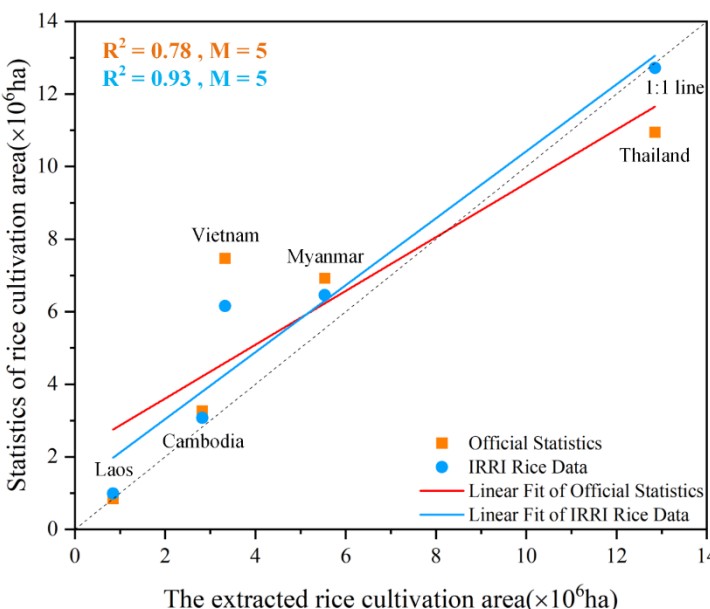

**Figure 7: Comparison of the extracted rice area with statistical rice harvested area and IRRI dataset at national level scale. M is the number of countries.**

## 4.3 Comparison with statistical data and other rice maps at the provincial scale

Figure 8 shows the comparison of the extracted rice area with the statistical data of rice harvested area and IRRI rice data at the provincial scale for five Southeast Asian countries. The available rice maps contain a 500 m resolution rice map of mainland Southeast Asia (IRRI Rice Data) and a 30 m resolution rice map of Vietnam (VLUCD) (see Sect. 2.2.3 for details). In general, the rice extraction results were in good agreement with the statistical area, IRRI data and VLUCD. Among them, the $R^2$ ranged from 0.82 to 0.88 with statistical data and from 0.83 to 0.97 with IRRI, as shown in Figure 8.

As shown in Figure 8 (a) and (b), the rice planting areas in Thailand and Cambodia extracted by our method had a good correlation with the statistical data and IRRI data at the provincial scale. The $R^2$ was distributed in the range of 0.83-0.88. There were no provinces with large deviations.

In Figure 8(c), in Myanmar, the $R^2$ between the extracted area of rice and the statistical data, IRRI rice data was 0.83 and 0.84, respectively. However, the extracted area of Ayeyarwady Province was significantly lower than that of the statistical data and IRRI data. The extraction results of Ayeyarwady were compared with the IRRI data, as shown in Figure 9. As reported by Han et al. (Han et al., 2021), due to the influence of mixed pixels, the IRRI data divides too many rivers and vegetation into rice. And the extracted rice map retains the details of rivers and roads.

The $R^2$ of the extracted rice area in Laos with statistical data was 0.82, and the highest agreement with IRRI data was 0.97, as shown in Figure 8(d). For the same reason as Ayeyarwady Province, the rice extraction area in Savannakhet Province was lower than the IRRI data because the details of rivers and roads were preserved in the extraction results.

Different from other sub figures, Vietnam added data comparison results with 30m VLUCD. The extraction results in Vietnam

355    correlated well with the statistical data, VLUCD and IRRI data, with all $R^2$ values greater than 0.80, as shown in Figure 8(e).

The area of rice extraction in Vietnam was in higher agreement with VLUCD ($R^2$ of 0.87) than with statistics ($R^2$ of 0.86) and

IRRI Rice Data ($R^2$ of 0.83). Most of the points of VLUCD were distributed on the 1:1 line.

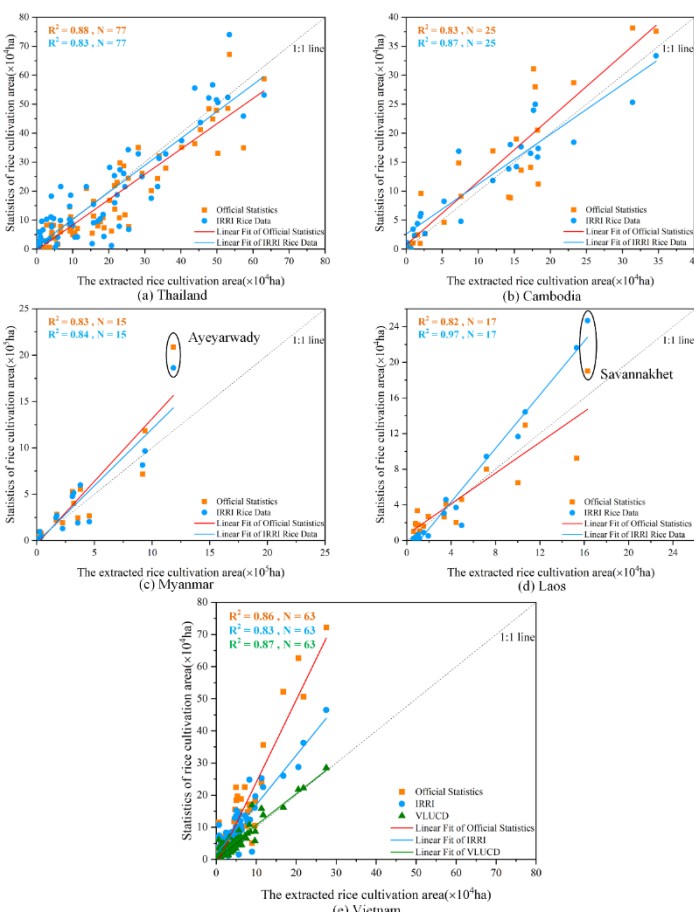

360    **Figure 8: Comparison of the extracted rice area with statistical rice harvested area and IRRI dataset at provincial scale. N is the number of provinces in each country.**





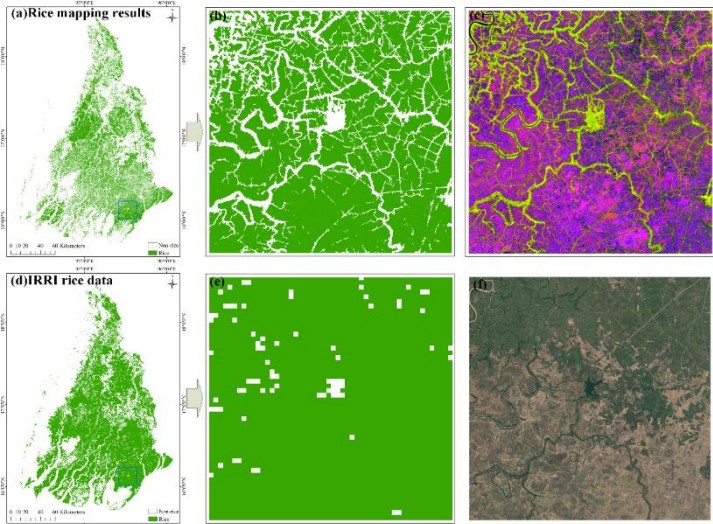

**Figure 9: Ayeyarwady's extracted rice area plotted against IRRI data. Our extraction results (a-b), False color image(c), IRRI data (d-e), Google Earth optical image (f) ©Google Earth.**

## 5 Discussion

In this study, annual rice maps for five Southeast Asian countries in 2019 were generated using temporal features extracted based on Sentinel-1SAR time series and an improved U-Net. Accuracy, Precision, and Recall based on the validation set exceeded 90% with a Kappa of 0.8425. Accuracy evaluation of rice mapping showed that the proposed temporal features were able to portray the unique growth characteristics of rice, and the improved U-Net model was able to suppress the false alarms of sporadic distribution caused by complex topography. The proposed method has superior capability in mapping rice distribution in large tropical regions.

The rice extraction results were compared with statistical data from the national and provincial levels in Sections 4.2 and 4.3. The results of multiple comparisons show that our rice extraction results are in high agreement with the statistical data. However, there were also minor inconsistencies. A possible reason is that the statistical cycle is not strictly aligned with the data collection cycle. The rice area extracted in this study is the total area of all fields that have been planted with rice in a year. Most agricultural statistics record the total area of rice planted in different growing seasons on an annual basis or even from one month of one year to the next. In addition, the statistical methods may cause errors in the statistics. The well-organized rice growing seasons were mainly considered in all statistics, and the random and irregular planting behavior of individual farmers was inevitably ignored. Considering the data collection conditions and statistical errors, it is understandable that the extracted rice maps differ from the official statistics.

The comparison results between rice area products extracted based on different remote sensing data show that our rice extraction results are in good agreement with the available rice products at the national and provincial levels. To fully





demonstrate the reliability of the rice extraction results, three subregions from the rice map were selected for comparison in Thailand and Vietnam, as shown in Figure 10. As mentioned in other literature (Dong et al., 2015; Han et al., 2021), the

MODIS-based IRRI rice map with 500 m resolution contains a large number of mixed image elements, and thus misclassification exists in the rice results. The spatial distribution characteristics of our rice map were generally consistent with those of the IRRI data, and our rice map retained more details with fewer mixed pixels. In addition, our rice map also had better agreement with the spatial distribution and detailed information of rice from VLUCD. Overall, comparisons based on the validation set, statistical data, and other rice products confirmed the reliability of our generated rice maps.

In the study, we found that the temporal features along rivers and wetlands are more similar to rice and have similar colors in the feature map, which can be easily misclassified as rice. The backscattered information of scattered rice fields is subject to interference from topography and surrounding non-rice land cover, resulting in missed detection. In future studies, corrections can be made using water masks extracted from higher precision land cover data.

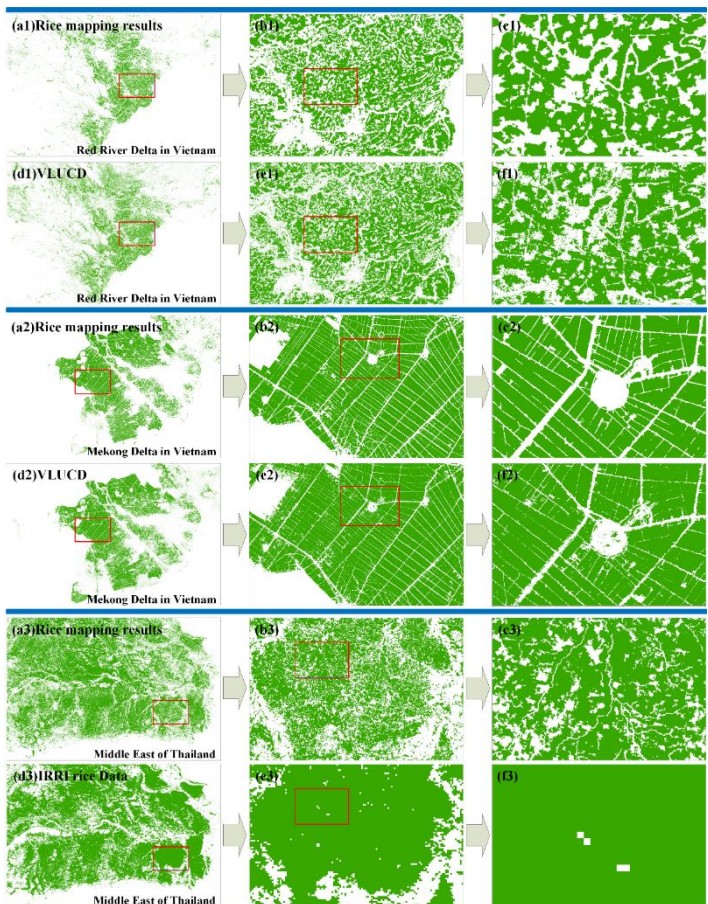


**Figure 10 Comparison of our rice maps with available products in typical regions. Our extraction results (a1-c1, a2-c2, a3-c3); VLUCD rice map (d1-f1, d2-f2); IRRI rice Data (d3-f3).**



## 6 Data Availability

The 20 m annual paddy rice map for mainland Southeast Asia can be accessed at Zenodo dataset from the following DOI:
https://doi.org/10.5281/zenodo.7315076(Sun, 2022). The spatial reference system of the dataset is EPSG:4326(WGS84).

## 7 Conclusions

Ending hunger and malnutrition in Southeast Asia is essential and rice plays a critical role. Rice is the single most important staple in the region as it provides 50% of calorie intake for its population. Satellite-based remote sensing offers the most practical means of monitoring changes on the vast area of land under rice cultivation in mainland Southeast Asia, given the
synoptic coverage, repeated and regular observation, and archival nature of satellite imagery. Questions remain, however, as to appropriate timing, number of satellite observations, spatial resolution of satellite imagery, and effective data processing methods for accurately capturing changes in factors such as rice production extent, growing seasons, and land management.
For large-scale rice mapping in tropical and subtropical regions, a rice mapping method based on time series SAR features and deep learning models is proposed. Rice mapping was completed for mainland Southeast Asia using the 2019 Sentinel-1 time
series data and the proposed rice extraction method. The accuracy of the proposed method on the validation sample set was 92.20%. Our rice maps correlated significantly with statistical data and were consistent with other rice maps. These results demonstrate the advantages of the proposed method for rice extraction with complex cropping patterns. The rice maps we produced will provide data support for agricultural resource studies, such as yield prediction and agricultural management.

**Author Contributions:** Conceptualization, methodology, software, C.S. and H.Z.; validation, formal analysis, H.Z.; investigation, C.S. and L.X.; resources, data curation, J.J. and J.G.; writing—original draft preparation, C.S. and H.Z.; writing—review and editing, H.Z., L.X., J.G. and L.Z.; visualization, L.X. and J.G.; supervision, project administration, H.Z. and C.W. All authors have read and agreed to the published version of the manuscript.

**Funding:** This research was funded by the National Natural Science Foundation of China under Grants 41971395, 41930110
and 42001278 and the Strategic Priority Research Program of Chinese Academy of Sciences (XDA19090119).
**Acknowledgments:** The authors would like to thank ESA and EU Copernicus Program for providing the Sentinel-1 SAR data.

**Conflicts of Interest:** The authors declare no conflict of interest.

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
