# Peer review of "m Annual Paddy Rice Area Map for Mainland Southeast Asia Using Sentinel-1 SAR Data"

_Earth System Science Data, 2022_

## Author Comment (AC1)

Dear reviewers                                                    Feb.20, 2023

Manuscript ID essd-2022-392 entitled

"20 m Annual Paddy Rice Map for Mainland Southeast Asia Using Sentinel-1 SAR Data".

We really appreciate the positive feedback of the editor and two referees and own many thanks to their reviews. We appreciate the thorough reviews provided by the editor and two referees again. We agree with these suggestions and have revised the manuscript accordingly. At the same time, in order to improve the quality of the paper, we also further modify the expression of the full text. Below is our response to his/her comments resulting in some clarification. We hope these revisions resolve the problems and uncertainties pointed out by the referee. In the manuscript and this file, the blue parts are revisions suggested by the reviewer 1, green parts for suggestions of reviewer 2. And the red parts are the changed contents that are intended to improve the expressions.

Sincerely,

Hong Zhang

zhanghong@radi.ac.cn

**Response to Reviewer 1**

**Comments to the Author**

**The manuscript proposes a rice mapping method to construct representative rice growth patterns based on time-series Sentinel-1 SAR data. Apparently, use of Sentinel-1 SAR data for rice mapping is very useful and seems the work is ongoing in the region for some time. Such methodology is useful, especially if it can give the rice area in advance which can help estimate the production helping in making timely trade/consumption plans. Below are few comments that should be helpful to improve the clarity and completeness of the manuscript.**

**RESPONSE:** Thank you very much for your appreciation of our work.

**1. What does this statement, .. rice growth patterns in Southeast Asia are too complex…means? Complexity refers to what? Is it not complex in other regions?**

**RESPONSE:** Thank you very much for your comment.

Sorry that the inaccurate word has troubled you. We referred to relevant literatures and changed the term "growth pattern" to "cultivation pattern" or "planning pattern" in the manuscript. We also revised the entire text.

The climate of Southeast Asia is suitable for the growth of rice all year round. There is no strict restriction on the planting time of rice. The planting time of rice in the whole region is not uniform, and the growing stages of rice in different regions are not synchronized. It is difficult to describe the cultivation of rice with a unified phenological information. Therefore, in this paper "complexity" refers to the diversity of rice planting patterns, especially for the entire land mass of Southeast Asian countries, which makes it difficult to establish a representative rice crop model. In this paper we attempt to establish a method for rice area extraction based on SAR data analysis, rather than relying more on ancillary data, such as complex phenological characteristics of various regions, or huge amounts of ground survey information.

Asia is the main rice producing region in the world. Take China, the world's leading rice producer, as an example. Affected by the climate, northern China usually grows one season of rice, while southern China usually grows two seasons of rice. Although the rice planting patterns in the north and south are different, for the same region the time of year when rice is grown is relatively concentrated (with some regulation by the local government), and it is easy to establish a representative rice crop model for rice area mapping.

**2. This is nice to have high-precision rice area mapping but why is it needed. Please explain for the benefit of readers in section 1. Also is it only high precision or high resolution too?**

**RESPONSE:** Thank you very much for your suggestion. We added the reason for the need for high-precision rice mapping in section 1.

We proposed an efficient rice area mapping method based on time series SAR features and a deep learning model to perform large-scale rice area mapping in tropical and subtropical regions. It is described in the manuscript that most of the rice area products covering Southeast Asia have a resolution of 500m. The 500 m resolution is still a bit of a gap for finer agricultural applications. And the high-precision rice area map we obtained has a spatial resolution of 20 meters. This resolution can meet the actual application requirements, and the data processing capacity is also moderate. In the future we plan to provide long-term rice area mapping products for this region.

**Page 1,line 29**

…High-precision rice planting area maps are the basis for monitoring rice growth and forecasting yields, the cornerstone for the government, planners and policymakers to formulate reasonable policies, and the guarantee of global food security (Mosleh et al., 2015; Laborte et al., 2017; Clauss et al., 2018; Jin et al., 2018; Yu et al., 2020; Hoang-Phi et al., 2021) …

**3. Were the data downloaded for whole year of 2019 of specific season? If so, the seasonal difference in rice area matters although there may be irrigation in some areas of some countries with 2 to 3 rice crops in irrigated areas and one crop in non-irrigated areas (as mentioned as Rice 1 through 4 in the manuscript). How was this seasonal difference considered in analysis in mapping the rice area? Because rice area per seasons will be different for the same country.**

**RESPONSE:** Thank you very much for your suggestion. All Sentinel-1 data for the whole year of 2019 were downloaded, but not divided by season. Our method extracted three effective temporal statistical features from the 2019 annual data, and then input them into the semantic segmentation model to obtain the 2019 rice area map. The proposed method directly obtains the information of whether the plots are planted with rice or not, without the need to segment the rice planting season, which is the advantage of our method. We will consider the seasonal difference of rice area in the subsequent study of multi-season rice area extraction.

**Table 1. List of Sentinel-1 SAR data in 2019 used in this study**

| Country | Satellite | Orbit-Frame | Number of images | Country | Satellite | Orbit-Frame | Number of images | Country | Satellite | Orbit-Frame | Number of images |
|---|---|---|---|---|---|---|---|---|---|---|---|
| Experimental Data | | | | | | | | | | | |
| Myanmar | S1A | 41-44 | 31 | Thailand | S1B | 62-1 | 29 | Vietnam | S1A | 55-31 | 31 |
| | | 41-50 | 31 | | | 62-2 | 29 | | | 55-37 | 31 |
| | | 41-55 | 31 | | | 62-3 | 29 | | | 55-42 | 31 |
| | | 41-60 | 31 | | | 62-4 | 29 | | | 55-47 | 31 |
| | | 41-65 | 31 | | | 62-5 | 29 | | | 55-62 | 31 |
| | | 41-70 | 31 | | S1A | 62-20 | 27 | | | 55-67 | 31 |
| | S1A | 70-1217 | 31 | | | 62-21 | 27 | | | 55-72 | 31 |
| | | 70-1222 | 31 | | | 62-22 | 26 | Laos | S1A | 26-44 | 31 |
| | | 70-1227 | 31 | | | 62-23 | 24 | | | 26-49 | 31 |
| | | 70-1232 | 31 | | | 62-24 | 25 | | | 26-54 | 31 |
| | | 70-1237 | 31 | | S1B | 91-1 | 32 | | | 26-59 | 31 |
| | | 70-1242 | 31 | | | 91-2 | 32 | | | 26-64 | 31 |
| | | 70-1247 | 31 | | | 91-3 | 32 | | | 26-69 | 31 |
| | | 70-1252 | 31 | Thailand | | 91-4 | 32 | | | 99-1240 | 30 |
| | | 70-1257 | 31 | | | 135-16 | 23 | | S1A | 99-1245 | 30 |
| | | 70-1262 | 31 | | S1A | 135-17 | 23 | | | 99-1250 | 30 |
| | | 70-1267 | 31 | | | 135-18 | 23 | | | 128-44 | 30 |
| | S1A | 143-46 | 30 | | | 135-19 | 23 | | S1A | 128-49 | 30 |
| | | 143-51 | 30 | | | 164-1 | 32 | | | 128-54 | 30 |
| | | 143-56 | 30 | | S1B | 164-2 | 32 | | | 128-59 | 30 |
| | | 143-61 | 30 | | | 164-3 | 32 | | | 128-64 | 30 |
| | | 143-66 | 30 | | | 164-4 | 32 | Cambodia | S1A | 26-29 | 28 |
| | | 143-71 | 30 | | | 164-5 | 32 | | S1B | 26-32 | 30 |
| | | 143-76 | 30 | | S1A | 164-20 | 13 | | | 26-38 | 30 |
| | S1A | 172-1248 | 28 | | S1A | 172-17 | 31 | | | 26-43 | 30 |
| | | 172-1253 | 28 | | | 172-18 | 31 | | S1A | 99-1220 | 30 |
| | | 172-1258 | 28 | Vietnam | S1A | 26-23 | 28 | | | 99-1225 | 30 |
| | | 172-1263 | 28 | | | 26-34 | 31 | | S1B | 99-31 | 31 |
| | | 172-1268 | 28 | | S1A | 128-29 | 30 | | S1A | 128-34 | 30 |
| | | 172-1273 | 28 | | | 128-69 | 30 | | | 128-39 | 30 |
| Training Dataset | | | | | | | | | | | |
| Thailand | S1A | 99-16 | 29 | | | | | | | | |

**4. Line 207-208, …the high heterogeneity of rice backscattering coefficients in Southeast Asia is caused by the high heterogeneity in climate and topography…what does mean that? Climate is obvious but what is heterogeneity in topography?**

**RESPONSE:**     Thank you very much for your comment.

First, climate influences rice cultivation patterns through precipitation and temperature. The spatial distribution corresponding to climate is large scale. Therefore, different topography within the geographical area covered by a climatic zone can also cause differences in temperature and temperature. The same can have an impact on rice cultivation patterns.

Second, our previous studies have shown that rice planted in relatively flat areas has a high degree of planting intensification, and the flooding period of the backscatter time series curve is more obvious. However, rice planted in areas with large topographic relief is easily affected by other non-rice plots due to the small and irregular plot area, and the flood period of the backscatter time series curve is not obvious. The sample production process also added samples from this region to ensure that the samples were representative.

**5. Figure 3 is not so nice. It should be improved to be more clear to read.**

**RESPONSE:** Thank you very much for your suggestion. We redraw Figure 3 to enhance its the readability. The other figures and tables have been modified in some way to give a clearer presentation.

**Page 10, line 230**

[Figure]

**Figure 3.** **The average VH polarization backscattering coefficient curve of typical landcovers (The shaded areas refer to the standard deviation calculated from the sample points).**

**6. Line 223, if all flooded, why rice will differ significantly from other crops as other crops will also be flooded and possible similar backscatter, unlike for example sugarcane field which certainly may have different backscatter as they may not submerged. Please clarify the statement – the other crops.**

**RESPONSE::** Thank you very much for your suggestion. Sorry for the ambiguity caused by inaccurate words. We revised this sentence. The flooding stage is a phase of rice growing stages.

**Page 11, line 242**

…During the flooding stage, the backscattering characteristics of rice are significantly different from other crops that do not require extensive irrigation, and are close to that of water.….

7. **Figure 4, the term 'Building' does not reflect that square. Appropriate name is settlement. Rice is also vegetation, so vegetation may better be called as non-rice vegetation. What is band combination for optical image?**

**RESPONSE:** Thank you very much for your suggestion. We modified the names of these two land covers in Figure 4 and the manuscript. We used the optical images of true color composite from Google Earth directly.

[Figure]

**Figure 4. The pseudo-color image synthesized from three SAR feature parameters (R: $\sigma_{max}^0$; G: $\sigma_{min}^0$; B: $\sigma_{var}^0$ ) and the corresponding optical image from Google Earth ©Google Earth.**

8. **How was number of (sample)plots of 1913 and 2032 for validation determined, any basis?**

**RESPONSE:** Thank you very much for your comment. The preparation method of rice samples, the distribution and quantity of plots have all referred to previous

studies(Singha et al., 2019; Wei et al., 2022; Lin et al., 2022), ensuring the accuracy and representativeness of plots and the sufficient number of plots. The distribution of the validation sample plots is shown in Figure 1.

[Figure]

**Figure 1. Location of the study area. The Sentinel-1 data with Orbit-frame 99-16 were used for the training samples, and the Rice and Non-rice flags show the distribution of the validation sample set. The base map is from Esri.**

**9. Country's statistics on area (despite may not align with data collection cycle) are most authentic. Table 5: the extracted rice area is only 44% of the statistics of rice cultivation area. This is rather huge difference. How to explain the feasibility of using the rice mapping method recommended by this study because of that discrepancy? It has now only Vietnam in the table. It is better to show all the countries data in the table.**

**RESPONSE:** Thank you very much for your suggestion.

The statistical data were the total rice harvest areas in different growing seasons each year, but the extracted rice area was the land area where rice was planted. Vietnam's statistical yearbook mentions that there are three seasons of rice, namely, spring rice, autumn rice, and winter rice, while the harvested areas of spring rice and

autumn rice are comparable, and the harvested area of winter rice is smaller. In this way, part of the statistical data of the rice harvest area is repeated and accounts for a large proportion of the area, resulting in a larger rice statistical area than the extracted rice area. Therefore, the extracted rice area of Vietnam is quite different from the statistical data. Although other countries also have multiple rice seasons, the areas of rice in the main season are large, while that in other seasons is small, so the area proportion calculated repeatedly is small.

It is noteworthy that the extracted rice area was closer to the paddy land area in the statistical yearbook of Vietnam and VLUCD (obtained by remote sensing), indicating that the extraction result was reliable and the proposed method is feasible.

According to your suggestions, we have supplemented the contents of Table 5.

**Page 17, line 363**

**Table 5. Statistics, other rice area maps and the extracted rice area for five Southeast Asian countries.**

| Country | Statistics of rice cultivation area ($\times 10^{\wedge 6}$ ha) | IRRI rice data ($\times 10^{\wedge 6}$ ha) | Statistics of paddy land area ($\times 10^{\wedge 6}$ ha) | VLUCD ($\times 10^{\wedge 6}$ ha) | Extracted rice cultivation area ($\times 10^{\wedge 6}$ ha) |
|---|---|---|---|---|---|
| Thailand | 10.9442 | 12.7198 | - | - | 12.8508 |
| Cambodia | 3.2638 | 3.0740 | - | - | 2.8215 |
| Myanmar | 6.9209 | 6.4575 | - | - | 5.5390 |
| Laos | 0.8435 | 0.9856 | - | - | 0.8458 |
| Vietnam | 7.4695 | 6.1527 | 4.1205 | 3.8210 | 3.3270 |

**10.    The method itself of rice mapping using Sentinel-1 SAR Data is major output of the study. If the proposed method is referring to Figure 2 flowchart, then, it still shows the need of using statistical data and available rice maps. Hence, better discussion with rationale is needed whether proposed method is to replace the existing system as efficient, accurate and even pre-harvest season rice mapping method or just additional task of mapping the rice area. Ideally, the new method is to replace or improve the existing method. It would be nice to indicate with the**

**proposed method whether existing statistical data collection on rice is still needed. Also, advise (recommend) how this new method can be made available for the country to use? Afterall the best use of method if adopted will be for the country.**

**RESPONSE:** Thank you very much for your suggestion.

Compared with the existing methods, our proposed method only uses Sentinel-1 SAR data as the data source, without using multi-source data. Sentinel-1 data is free and has a high temporal and spatial resolution. Moreover, the proposed method is simpler and more efficient. The high-precision rice area mapping can be completed using the time series statistical features and the classic semantic segmentation model.

The rice area map we obtained is evaluated from multiple perspectives by validation samples, rice statistical data, and other rice area products. Comparing rice area extraction results with statistical data is a standard accuracy verification method (Han et al., 2021; Wei et al., 2022; Son et al., 2022). In the future, after the improvement of the method and continuous validation, a very reliable rice area extraction system will be obtained, which may not require statistical data.

We will actively and continuously produce rice area data from 2019 onwards. This rice area data will be released through the International Research Center of Big Data for Sustainable Development Goals (CBAS)for free use by the state and the public.

**References**

Han, J., Zhang, Z., Luo, Y., Cao, J., Zhang, L., Cheng, F., Zhuang, H., Zhang, J., and Tao, F.: NESEA-Rice10: high-resolution annual paddy rice maps for Northeast and Southeast Asia from 2017 to 2019, Earth System Science Data, 13, 5969-5986, 10.5194/essd-13-5969-2021, 2021.

Lin, C., Zhong, L., Song, X.-P., Dong, J., Lobell, D. B., and Jin, Z.: Early-and in-season crop type mapping without current-year ground truth: Generating labels from historical information via a topology-based approach, Remote Sensing of Environment, 274, 112994, https://doi.org/10.1016/j.rse.2022.112994, 2022.

Singha, M., Dong, J., Zhang, G., and Xiao, X.: High resolution paddy rice maps in cloud-prone Bangladesh and Northeast India using Sentinel-1 data, Sci Data, 6, 26,

10.1038/s41597-019-0036-3, 2019.

Son, N.-T., Chen, C.-F., Chen, C.-R., Cheng, Y.-S., Toscano, P., Syu, C.-H., Guo, H.-Y., Chen, S.-L., Liu, T.-S., and Zhang, Y.-T.: Exploiting Sentinel-1 data and machine learning–based random forest for collectively mapping rice fields in Taiwan, Applied Geomatics, 1-15, 10.1007/s12518-022-00440-4, 2022.

Wei, J., Cui, Y., Luo, W., and Luo, Y.: Mapping Paddy Rice Distribution and Cropping Intensity in China from 2014 to 2019 with Landsat Images, Effective Flood Signals, and Google Earth Engine, Remote Sensing, 14, 759, 10.3390/rs14030759, 2022.

---

## Author Comment (AC2)

Dear reviewers                                         Feb.20, 2023

Manuscript ID essd-2022-392 entitled

"20 m Annual Paddy Rice Map for Mainland Southeast Asia Using Sentinel-1 SAR

Data".

We really appreciate the positive feedback of the editor and two referees and own many thanks to their reviews. We appreciate the thorough reviews provided by the editor and two referees again. We agree with these suggestions and have revised the manuscript accordingly. At the same time, in order to improve the quality of the paper, we also further modify the expression of the full text. Below is our response to his/her comments resulting in some clarification. We hope these revisions resolve the problems and uncertainties pointed out by the referee. In the manuscript and this file, the blue parts are revisions suggested by the reviewer 1, green parts for suggestions of reviewer 2. And the red parts are the changed contents that are intended to improve the expressions.

Sincerely,

Hong Zhang

zhanghong@radi.ac.cn

**Response to Reviewer 2**

**Comments to the Author**

**As rice land mapping is significant in Northeast Asia, the authors developed a mapping method based on time-series Sentinel-1 SAR data in this area. Based on the analysis of rice backscattering characteristics in the aimed area. Based on the analysis of rice backscattering characteristics, they combined spatio-temporal statistical features with the generalization ability to rice growth patterns, together with the adoped model approaches, the 20-meter resolution rice land cover map of five countries in mainland Southeast Asia in 2019 was obtained. The methods are helpful for agricultural management in the related areas. Similar methods can be adopted to make a regional rice cover map by repeating the experiment in this paper according to subsequent data. However, I have the following suggestions or concerns that maybe helpful for the modification of the manuscript.**

**RESPONSE:** Thank you very much for your appreciation of our work. Your suggestions have helped us a lot to improve the quality of our manuscript.

My major concerns are:

**1. Is the title accurate? It is strange to call rice mapping. I would like to suggest you take into careful consideration for this. Rice land mapping? Rice land cover mapping. Anyway, rice mapping is not so suitable.**

**RESPONSE:** Thank you very much for your suggestion.

According to your suggestion, we have made statistics on the published papers from 2017 to now (See end of the response for details). There are eight kinds of titles commonly used for extracting the area of rice plots namely "Mapping Paddy Rice", "Rice Mapping", "Mapping of Paddy Rice Extent", "Mapping Rice Fields", "Rice Planting Areas", "Detecting Rice Paddy" and "Mapping of Paddy Rice Extent".

Specifically, there are 11 papers with the title of mapping paddy rice(Wang et al., 2022; Yang et al., 2021b; Xu et al., 2021; Ni et al., 2021; Han et al., 2021; Wang et al.,

2020; Li et al., 2020; Bazzi et al., 2019; Song et al., 2018; Park et al., 2018; Lasko et al., 2018).

There are 9 papers with the title of rice map(Liu et al., 2022; Lin et al., 2022; Zhan et al., 2021; Yang et al., 2021a; Wei et al., 2021; Singha et al., 2019; Guo et al., 2019; Talema and Hailu, 2020; Zhao et al., 2021).

There are 4 papers with the title of mapping of paddy rice extent(Soh et al., 2022; Rudiyanto et al., 2019; Zhang et al., 2018; Tian et al., 2018).

There are 3 papers with the title of mapping rice fields(Son et al., 2022; Chen et al., 2020; Chang et al., 2020b).

There are 2 papers with the title of rice planting areas(Hoang-Phi et al., 2020; Stroppiana et al., 2019).

There are 2 papers with the title of detecting rice paddy(Jo et al., 2020; Crisóstomo De Castro Filho et al., 2020).

There are 2 papers with the title of mapping rice area(Stroppiana et al., 2019; Clauss et al., 2017).

There are 2 papers with the title of mapping paddy rice distribution(Wei et al., 2022; Chang et al., 2020a).

For more accurate expression, we changed the title to "20 m Annual Paddy Rice Area Map for Mainland Southeast Asia Using Sentinel-1 SAR Data". In addition, we also carefully examined the full manuscript and changed the "rice map" to "rice area map" to ensure a clear and accurate expression.

**2. Section 3.2: feature extraction was not told very clearly, and affected understanding for the whole processing process for mapping. In particular, the comparison between pseudocolor image and optical image (figure 4), and the role of the optical image in cartography were not clarified.**

**RESPONSE:** Thank you very much for your suggestion.

In section 3.2 feature extraction, firstly, the variation characteristics of VH polarization time series backscatter coefficients of four typical land cover types (rice, water, settlement and non-rice vegetation) in the study area were analyzed. It can be

seen that the time steps of each growing season for the selected Rice 1- Rice 4 were inconsistent. This makes the backscatter coefficient curves of the rice growth cycle more diverse and does not allow us to summarize a generalized model of rice evolution. Therefore, it will be difficult to accomplish the rice extraction task using a direct reliance on the fixed relationship between phenology and time.

Through a large number of comparative experiments and analysis, and in combination with our previous research work (Xu et al., 2021), three time series statistical features were selected to describe the most significant SAR features in the rice growth process for rice mapping in the study area. Finally, the pseudo-color image synthesized by three time-series statistical features is compared with the optical image to show that rice and other landcovers in the feature map have obvious differentiation.

The optical image in Figure 4 is mainly used for reference. In the pseudo-color feature map, rice has significantly different characteristics from other landcovers s (water, settlements and non-rice vegetation). In order to prove the accuracy of the landcovers selected, we selected the corresponding optical image for comparison.

The optical image plays an auxiliary role in the process of rice mapping, and is not used as the input of rice extraction model together with pseudo-color feature map. First of all, optical image was used to judge landcovers to assist in sample preparation in 3.3: training and validation sets. Secondly, optical image was used as the contrast of landcovers in the feature map in section 3.2: feature extraction. Finally, the rice extraction results were compared with optical images to illustrate the accuracy of the results in section 4: results.

I have also some minor points as the following:

**1. Abstract section: I am wondering what was the complex growth patterns for rice in tropical and subtropical regions. Since this is too vagur, would you please tell in details?**

**RESPONSE:** Thank you very much for your suggestion. Sorry that the inaccurate word has troubled you. We referred to relevant literature and changed the "growth pattern" to "cultivation pattern" or "planning pattern" in the manuscript. The rice planting pattern

(or cultivation pattern) is a planting attempt on land that regulates the layout and sequence of planting for a certain period. For example,there are three patterns of rice planting pattern:(1) paddy-paddy-paddy, (2) paddy-paddy-fallow, (3) paddy-fallow-paddy.

The climate of Southeast Asia is suitable for the growth of rice all year round. There is no strict restriction on the planting time of rice. The planting time of rice in the whole region is not uniform, and the growing stages of rice in different regions are not synchronized. It is difficult to describe the cultivation of rice with a unified phenological information. Therefore, in this paper "complexity" refers to the diversity of rice planting patterns, especially for the entire land mass of Southeast Asian countries, which makes it difficult to establish a representative rice crop pattern. In this paper we attempt to establish a method for rice area extraction based on SAR data analysis, rather than relying more on ancillary data, such as complex phenological characteristics of various regions, or huge amounts of ground survey information.

**2. Introduction section: United Nation's sustainable development goals and the significances of food are not the most important for this manuscript, so please have some cut at this to make it short, and focus on the mapping methods and the advances in this field, and be noted that this is data manuscript.**

**RESPONSE**:Thank you very much for your suggestion. We have adjusted this part according to your suggestion to make it more concise.

**Page 1,line 25**

Sustainable development Goal 2 "Zero Hunger" was set by the United Nations in 2015(Desa, 2016). The dual pressure of population and environment threatens the sustainability of global food security (Faostat, 2010; Godfray et al., 2010). Rice feeds more than half of the world's population as a staple food and is a major crop for world food security (Kuenzer and Knauer, 2012). Asia is the largest rice-producing region in the world (Chen et al., 2012), and Southeast Asia accounts for 40% of global rice exports (Yuan et al., 2022). High-precision rice planting area maps are the basis for

monitoring rice growth and forecasting yields, the cornerstone for the government, planners and policymakers to formulate reasonable policies, and the guarantee of global food security (Mosleh et al., 2015; Laborte et al., 2017; Clauss et al., 2018; Jin et al., 2018; Yu et al., 2020; Hoang-Phi et al., 2021).

**3. Line 37, line 40-41, and also line 45-47: references were in wrong format or not suitable. Please also check many other places expressed like this.**

**RESPONSE:** Thank you very much for your suggestion. We added the missing part of the reference in line 37, deleted one reference in line 40-41, and adjusted the position of one reference in line 45-47 in the original manuscript. Moreover, we also checked and revised the references to ensure correct formatting and complete content.

**Page 2, line 41**

…Luo et al. and Wei et al. used Landsat time-series data to produce 1 km and 30 m resolution rice datasets for China, respectively (Luo et al., 2020; Wei et al., 2022)….

**Page 2, line 46**

…Xiao et al., Gumma et al. and Bridhikitti et al. produced low- and medium-resolution rice area maps for several South and Southeast Asian countries using MODIS data at the 500 m spatial resolution, respectively (Xiao et al., 2006; Gumma et al., 2011a; Gumma et al., 2011b; Bridhikitti and Overcamp, 2012; Gumma et al., 2014). Nelson and Gumma extracted the 500 m spatial resolution general rice extent map in Asia from 2000 to 2012 using MODIS data (Nelson and Gumma, 2015) ….

**4. Line 68-70: what was the difference between the two categories? The information in the second category can also include phenological information. So, confusing as far as this expression is concerned.**

**RESPONSE:** Thank you very much for your comment.

Although both methods may have phenological information, the role of phenological information is different. In the first method, phenological information

plays a major role. It is necessary to manually use the growing phenological calendar to identify the phenological periods of rice in the time series data or to make phenological indicators based on phenological information.

The second method is to directly input the time series data into the machine learning model, relying on the model to learn the rice information in the time series data, while the phenology information only plays an auxiliary role in the early stage of the study. Compared with the phenology-based method, this method is less dependent on human.

**5. Line 70-74: Please refer to relevant literature, the expression of crop phenology should be regulated.**

**RESPONSE**:Thank you very much for your suggestion. We use more accurate words for better reading.

**Page 3,line 74**

…The phenology-based approach refers to the extraction of rice by defining phenological indicators or identifying rice growing stages by combining the time-series data covering the rice growth cycle and the analysis of rice phenological calendar (Nelson et al., 2014; Chen et al., 2016; Nguyen and Wagner, 2017; Liu et al., 2018; Xin et al., 2020; Ni et al., 2021). The growing stages such as transplanting, heading and maturity are most often used to extract rice. Shew et al. combined vegetation indices extracted from Landsat time-series data with a rule-based algorithm for growing stages to map a 30 m dry season rice map of Bangladesh from 2014 to 2018 (Shew and Ghosh, 2019) ….

**6. Line 106: the editing was careless: see line 106 "inSoutheast Asia".**

**RESPONSE**:Thank you very much for your suggestion. We corrected this editorial error.

**Page 4, line 111**

Therefore, in this study, to meet the requirements of high-precision rice area mapping in Southeast Asia....

**7. Due to the limited information in table 1, I suggest make it supplement.**

**RESPONSE:** Thank you very much for your suggestion. We adjusted table 1 to shorten the space it occupies. Other tables have been revised to give a more attractive and concise presentation

**Page 6, line 163**

**Table 1. List of Sentinel-1 SAR data in 2019 used in this study**

[revised manuscript text omitted]

**8. Section 2.2.2: the agricultural statistical yearbooks may have great uncertainty, which may affect its credibility. Please discuss uncertainties it brought.**

**RESPONSE:** Thank you very much for your suggestion. We discussed the uncertainty of statistical data in the second paragraph of section 5 of the manuscript. In addition to statistical data, we also evaluate the precision of rice extraction results from various rice products at the national and provincial levels to prove the accuracy of the extraction results as much as possible.

…A possible reason is that the statistical cycle is not strictly aligned with the data collection cycle. The rice area extracted in this study is the total area of all fields that have been planted with rice in a year. Most agricultural statistics record the total area of rice planted in different growing seasons on an annual basis or even from one month of one year to the next. In addition, the statistical methods may cause errors in the statistics. The well-organized rice growing seasons were mainly considered in all statistics, and the random and irregular planting behavior of individual farmers was inevitably ignored. Considering the data collection conditions and statistical errors, it is understandable that the extracted rice maps differ from the official statistics...

**9. Section 3.1: it was told the Sentinel-1 time-series data were preprocessed by using SNAP software, but you omitted the mechanism for conducting such events, and did not mention the how to determine the authority of the software.**

**RESPONSE:** Thank you very much for your suggestion. We have explained this section in detail.

The Sentinle-1 time-series data were preprocessed using the Sentinel Application Platform (SNAP) (Filipponi, 2019). The SNAP is a common architecture for all Sentinel Toolboxes. ESA/ESRIN is providing the SNAP user tool free of charge to the Earth Observation Community.

The steps were as follows: (1) Orbit correction: This operation refines the inaccurate orbit state vectors provided in the metadata of a SAR product with the precise orbit files which are available days to weeks after the generation of the product;(2) Thermal noise removal: Because SAR are contaminated by additive thermal noise, this step is introduced to mitigate thermal noise effects;(3) Radiometric calibration: This process provides the image in which the pixel values can be directly related to the radar backscatter of the image;(4) Coregistration: This step co-registers multitemporal intensity images; (5) Terrain correction: This process converts SAR data from the slant or ground range projection to geographic coordinate projection and corrects the distortion effects that occurred during the acquisition (overlay, shading);(6) Multitemporal speckle noise filtering: This operation reduces speckles, which degrade the quality of the image and make interpretation of features more difficult;(7) Converting values to decibels: This step convers the multitemporal intensity map to sigma 0 ($\sigma^0$) on the decibel (dB) scale using a logarithmic transformation. The final $\sigma^0$ images with 20 m resolution in the WGS84 geographic coordinate system were obtained.

**10. Section 3.4: similar to above problem, the reason for choosing U-Net model was not introduced and affected the reliability of your processing results.**

**RESPONSE:** Thank you very much for your suggestion. We modified section 3.4 and Figure 5.

**Page 13, line 274**

In this paper,high-precision rice area mapping was accomplished using U-Net model. U-Net is a classical semantic segmentation model widely used in biomedical image segmentation and remote sensing (Wei et al., 2019; Xu et al., 2021; Lin et al., 2022). It outputs semantically labeled pixel-by-pixel images corresponding to the input image while extracting high-level semantic features, so that the spatial details of the input image can be maintained (Ronneberger et al., 2015). SAR images cover a large spatial area, which include multiple ground objects with complex and rich semantic

information; rice fields are spatially characterized by continuous and large distribution. Therefore, U-Net is used to fully combine the spatial and semantic information in SAR images to achieve high-precision rice area extraction.

The structure of U-Net model is shown in Figure 5. U-Net consists of encoder (contracting path) and decoder (expansive path). The encoder is used for feature extraction, and the decoder is used to restore the size of the input image. U-Net has 23 convolutional layers, including eighteen $3 \times 3$ convolutional layers, four $2 \times 2$ convolutional layers and one $1 \times 1$ convolutional layer. The encoder part consists of five downsampling units, where each unit consists of two $3 \times 3$ convolutional layers and a $2 \times 2$ max-pool layer. The output of the downsampling unit is input to the next downsampling unit by max-pooling. The decoder contains four upsampling units, each of which consists of two $3 \times 3$ convolutional layers and a $2 \times 2$ deconvolutional layer. In the final stage of the decoder, the feature vector of the last upsampling unitis converted into a probability mapping by the $1 \times 1$ convolutional layer. The dimension of the probability mapping is 2 and the pixel value indicates the probability that the pixel belongs to rice and non-rice.

Meanwhile, thanks to the U-shaped structure and skip connection, each downsampling is cascaded with the corresponding upsampling, and this fusion of features at different scales is greatly helpful for upsampling to recover pixels. Specifically, the shallow downsampling multiplier is small and the feature map has more detailed rice spatial distribution features (low-level spatial features). While the deep downsampling multiplier is large and the information is heavily condensed with large spatial loss. But the high-level semantic features obtained from deep downsampling help in the determination of rice regions. When the high-level and low-level features are fused, it helps to improve the segmentation accuracy.

To solve the problem of uneven data distribution, we added a batch normalization (BN) layer (Ioffe and Szegedy, 2015) before each convolutional layer. The BN layer allows the input data to follow the same distribution to achieve regularization of the model.

[Figure]

Figure 5. Structure of the U-Net model.

**11. Section 7, the conclusion section: the expression were not suitable, since you should tell the most important findings for the rice land mapping technology in this work, but not the evaluation and remark on that.**

**RESPONSE:** Thank you very much for your suggestion. We revised the conclusion according to your suggestion.

**Page 22, line 440**

Ending hunger and malnutrition is essential, and rice plays a critical role. Satellite-based remote sensing offers the most practical means of monitoring rice cultivation in mainland Southeast Asia. Questions remain, however, as to appropriate timing, number of satellite observations, spatial resolution of satellite imagery, and effective data processing methods for rice distribution and production information.

To perform large-scale rice area mapping in tropical and subtropical regions, an efficient rice area mapping method based on time series SAR features and a deep learning model is proposed. A 20-meter spatial resolution rice area map of mainland Southeast Asia was produced using the 2019 Sentinel-1 time series data and the proposed rice area mapping method. The accuracy of the proposed method on the validation sample set was 92.20%. Our rice area maps correlated significantly with

statistical data and were consistent with other rice area maps. These results demonstrate the advantages of the proposed method for rice area mapping with complex cropping patterns. The rice maps we produced will provide data support for agricultural resource studies, such as yield prediction and agricultural management.